# Anticancer Properties of Aqueous Extracts from Leguminosae

Luca Serventi [1,*], Xuanyi Cai [1,2], Ruitian Chen [1,2], Nadeesha Dilrukshi [3], Jingyi Su [1], Refi Priskila Novaleta Tuange [1] and Elizabeth Eilidh Ham [1]

1   Department of Wine, Food and Molecular Biosciences, Lincoln University, Christchurch 7647, New Zealand
2   College of Light Industry and Food, Academy of Contemporary Agricultural Engineering Innovations, Zhongkai University of Agriculture and Engineering, Guangzhou 510225, China
3   Department of Livestock and Avian Sciences, Faculty of Livestock, Fisheries and Nutrition, Wayamba University of Sri Lanka, Makandura 60170, Sri Lanka
*   Correspondence: luca.serventi@lincoln.ac.nz

**Abstract:** Inflammation and cancer are diseases caused by genetic and environmental factors as well as altered microbiota. Diet plays a role, with leguminous such as beans (*Phaseolus vulgaris*, *Vicia faba*), chickpeas (*Cicer arietinum*), lentils (*Lens culinaris*), peas (*Pisum sativum*) and soybeans (*Glycine max*), known to prevent such diseases. Processing of food leguminous yields aqueous side streams. These products are nothing short of water extracts of leguminous, containing albumin, globulin, saponins, and oligosaccharides. This review analysed the most recent findings on the anticancer activities of legume-soluble nutrients. Albumin from chickpeas and peas inhibits the pro-inflammatory mediator interleukins, while soy Bowman–Birk Inhibitor inhibits serine proteases. The peptide vicilin activates peroxisome proliferator-activated receptor, mediating triglyceride metabolism. Soyasaponins promote apoptosis of cancer cells by activating caspases and by enhancing the concentration of intracellular calcium. Soyasapogenol regulates specific protein pathways, leading to apoptosis. Oligosaccharides such as raffinose and stachyose promote the synthesis of short chain fatty acids, balancing the intestinal microbiota, as result of their prebiotic activity. Verbascoside also modulate signalling pathways, leading to apoptosis. In closing, water extracts of leguminous have the potential to be efficient anticancer ingredients, by means of numerous mechanisms based on the raw material and the process.

**Keywords:** albumin; BBI; cancer; leguminous; oligosaccharides; saponins

## 1. Introduction

Inflammation involves a complex biological immune mechanism process that causes the release of pro-inflammatory cytokines and mediators such as interleukin (IL) 1β, IL-6, and tumour necrosis factor α (TNF-α) [1]. This is due to genetic factors, altered intestinal microbiota, environmental factors, and immunological abnormalities. In general, inflammatory diseases may be classified as acute or chronic, depending on the symptoms. Acute diseases are normally found as a natural reaction of the immune system to infection or body allergies. On the contrary, chronic diseases such as tumours and cancer are associated with chronic inflammatory conditions. The intestinal epithelial cells require cytokines as the key mediator in both physiology and pathophysiology [2,3]. Cytokine-guided interactions between epithelial and immune cells are pivotal to maintaining intestinal homeostasis. Consequently, studies about the tumour mechanism can be approached in two ways.

Bioactives found in leguminous include saponins, phenolics, vitamins, lipids and peptides. Sources are beans, chickpeas, lentils, peas and soy [4]. Diets rich in prebiotic carbohydrates lowered inflammatory bowel disease (IBD). Oligosaccharides found in leguminous and, likely, in legume water, were demonstrated to favour the growth of health-promoting microorganisms, resulting in the synthesis of SCFA [5].

Legume consumption requires processing: soaking, boiling, blanching, germination, fermentation and other techniques. Cooking leguminous also produces wastewater which has ignited interest in the food industry as a novel food ingredient. This side stream has been shown to contain numerous soluble nutrients: proteins (such as albumin and globulin), phenolics, saponins and carbohydrates (oligosaccharides and soluble fibre). For example, aquafaba (chickpea cooking water) contains approximately 1.5 g/100 mL of protein, 1.5 g/100 mL carbohydrates, 13 mg/100 mL of saponins and 0.7 mg/100 mL of phenolic compounds [6,7]. Water extracts of chickpeas have been recently shown to induce apoptosis in adenocarcinoma cells in vivo, activity attributed to polyphenols [8]. In addition, compounds found in aqueous extracts of Leguminosae have been shown to exert numerous bioactivities, such as anti-inflammatory, antioxidant, antiproliferative and antimicrobial [9–18] (Table 1).

**Table 1.** Key bioactives from aqueous extracts of Leguminosae and their mechanism of action.

| Nutrient | Source | Bioactivity | Cell Type | References |
|---|---|---|---|---|
| Albumin | *Vigna radiata* (Mung beans) | Antioxidant | ABTS test ORAC test | [9] |
| Defensin | *Lens culinaris* (Lentils) | Antifungal, anti-inflammatory | Caco-2 cells ATCC HTB-37 | [10] |
| Lunasin protease inhibitor (Lunasin, Kunitz, Bowman–Birk) | *Glycine max* (Soybeans) | Anti-inflammatory Antiproliferation of cytokines | RAW 264.7 | [11] |
| Vicilin | *Vicia faba* (Faba beans) | Anti-inflammatory | C2C12 | [12] |
| Oligosaccharides | Numerous sources | Prebiotic | Human studies | [13] |
| Saponins | *Glycine max* (Soybeans) | Anti-inflammatory | Murine alveolar macrophages line MH-S | [14] |
| | *Cicer arietinum* (Chickpeas) | Anti-proliferative | Human studies | [15] |
| Phenolics | *Phaseolus vulgari* (Cranberry beans) | Anti-inflammatory Antioxidant | Caco-2 cells | [16] |
| | *Cicer arietinum* (Chickpeas) | Anti-inflammatory Antioxidant | RAW 264.7 | [17] |
| | *Vigna radiata* (Mung beans) | Antioxidant | ABTS test | [18] |

It is hypothesised that water extracts of leguminous may exert anticancer activities, due to their content of albumin, globulin, saponins and prebiotic oligosaccharides. Therefore, this review critically analyses the most recent literature on cancer preventing mechanisms of soluble nutrients from leguminous.

## 2. Discussion

### 2.1. Soluble Proteins from Leguminous and Anticancer Mechanisms

2.1.1. Albumin

The albumin fraction represents a quarter of the protein content in chickpeas (*Cicer arietinum*) [19]. Comparatively, albumin in pea (*Pisum sativum*) consists of 15–25% of the total protein extracted [20]. Studies have shown the anti-inflammatory effects of albumin, leading to apoptosis of cancer cells [21,22]. The level of nitric oxide (NO) was compared between sprouted and non-sprouted chickpeas using an inflammation-activated macrophage RAW 264.7 (Table 2; Figure 1). Results showed an insignificant reduction of NO levels when using raw and cooked chickpea native protein. However, the protein digestion of cooked chickpeas and raw chickpeas was found to be five times more effective in inhibiting NO

production, showing the potential anti-inflammatory of chickpea albumin [21]. Another study investigated the effect of albumin extract from peas against the inflammation of mouse colitis [22]. Colitis was induced by adding dextran sulfate sodium to drinking water over the course of four days. The colitic mice treated with the aqueous pea extract showed a reduced mRNA expression of cytokines TNF-α, IFN-γ, IL-6 and IL-12. Despite studies conducted that have investigated the potential anti-inflammatory property of albumin protein [22], investigation of other bioactive components such as the Bowman–Birk inhibitor (BBI), antioxidants flavones cannot be excluded.

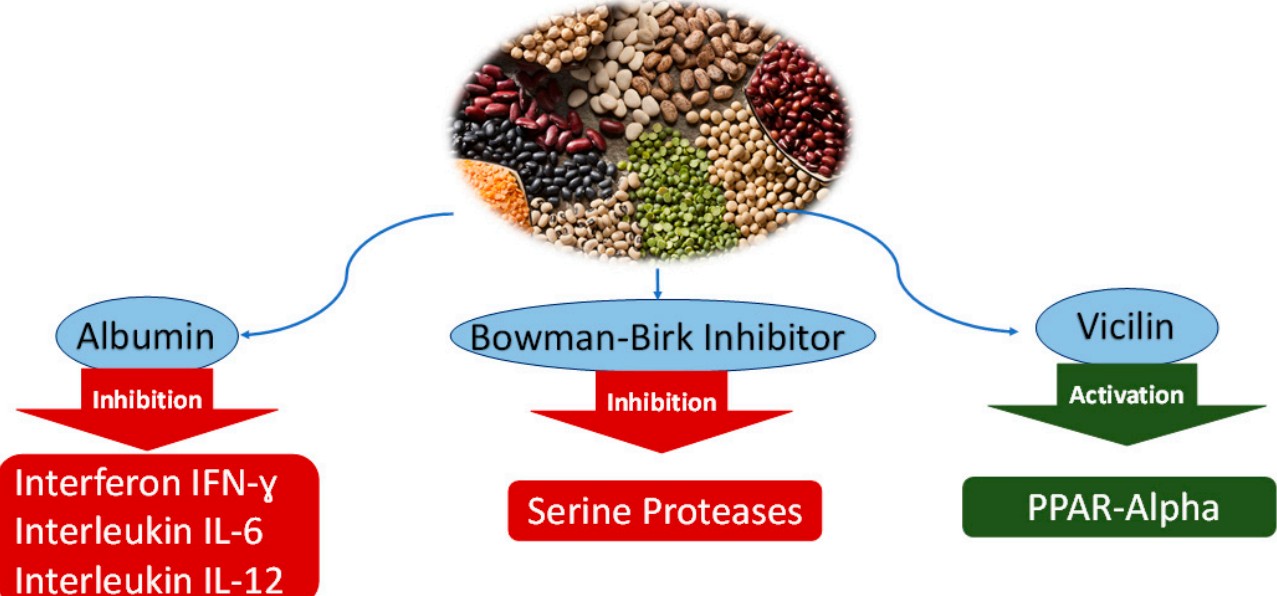

**Figure 1. Anticancer mechanisms of soluble protein from water extract of Leguminosae.** Inhibition of interleukins and serine proteases by albumin and Bowman–Birk Inhibitor, activation of PPAR-α by vicilin.

2.1.2. Bowman-Birk Inhibitor

Generally found in various leguminous, Bowman–Birk Inhibitor (BBI) is investigated to remain active reaching the large intestine due to its composition not being altered by gastric acids or other enzymes. The use of BBI from soy (*Glycine max*) was studied to significantly modulate the upregulated colonic mRNA expression of cytokines interferon IFN-γ, interleukins IL-6, and IL-12 in cells HT29 [22]. Its ability to inhibit serine proteases contribute to its anti-inflammatory property (Table 2; Figure 1). In addition to this, BBI has also been studied to have antiproliferative effects on human colon cancer cells [23]. An experiment was performed to investigate the potential effect of albumin extract found in chickpeas on inflammation using the DSS model of mouse colitis. In this experiment, soy BBI was found to be a promising substance to treat inflammation comparatively effective to the albumin fraction found in pea extract.

2.1.3. Vicilin

Globulins are salt soluble, representing 50–60% of the total protein in peas. Globulin plays a vital role in transporting nutrients and fighting infections. Globulins are mostly made by the liver and aid in carrying proteins, enzymes, and antibodies [24]. The globulin fraction is divided into two major groups, legumin and vicilin, characterized by their sedimentation coefficients. Often described as an antinutrient due to its allergic properties, the effect of vicilin on lipid metabolism was investigated. Anticancer properties were observed in numerous studies [24–27]. The effect of vicilin extracted from mung bean was investigated against the HMG CoAr enzyme in 3T3-L1 cells [25]. The study

shows potential peptides derived from mung bean (*Vigna radiata*) may contribute to the beneficial anticholesterolemic activity. Another investigation confirmed the vicilin effect in resolving lipid metabolism [26]. Vicilin hydrolysed extract was obtained from peas upon gastrointestinal simulation and resulted in modulated mRNA expression by exhibiting peroxisome proliferator-activated receptor (PPAR) $\gamma$ ligand activity, a promising result in the benefit of vicilin in leguminous to treat obesity-associated metabolic orders (Table 2; Figure 1). Activation of PPAR-$\alpha$ is reported to reduce triglyceride levels and is involved in the regulation of energy homeostasis while PPAR-$\beta$/$\gamma$ was reported to enhance fatty acids metabolism [26]. These properties then contribute towards the anticholesterolemic property of the vicilin protein found in mung beans. To this date, the author finds insufficient information regarding the use of particularly legumin protein extracted to combat inflammatory diseases. However, the previous study investigating the inhibition of nitric oxide using extracted proteins from germinated chickpeas shows promising results in legume protein extracts in combating inflammation.

*2.2. Saponins from Leguminous and Anticancer Mechanisms*

2.2.1. Saponins

Saponins are phytochemicals characterised by a non-polar alginate structure and polar sugar side chain [28]. Based on the differences in main structures, sapogenol (aglycones) of legume saponins can be divided into two categories: A and B. Soyasapogenol is conjugated with the DDMP (2,3-dihydro-2,5-dihydroxy-6-methyl-4H-pyrane-4-ketone) at the specific position of the structure in glycoside [28]. Therefore, legume saponins also are classified into A, B and DDMP types. The main saponins found in leguminous are group B saponins. They contain only one oligosaccharide chain, which can be divided into soyasaponin Ba (S1 saponins) and soyasaponin Bb (SV saponins) according to their structure. They are at the C-3 position, which is linked to the B portion of legume saponins. Moreover, soyasaponin Ba and soyasaponin Bb conjugated with DDMP on C-22 are called conjugated saponins $\alpha$-G (soyasaponin $\alpha$g-DDMP) and $\beta$-G (soyasaponin $\beta$g-DDMP) structures. It was found that $\beta$-G and S1 structures were similar in chickpea (*Cicer arietinum*), pea (*Pisum sativum*) and faba (*Vicia faba*), and slightly higher in SV and $\alpha$-G saponins [28].

Saponins have been shown to inhibit the proliferation of colon cancer cells [29–34] (Table 2; Figure 2). Additionally, saponins have also been suggested that they play pivotal roles in anti-inflammatory, cholesterol-lowering and immune-stimulatory [25]. They also protect cells from damage by reducing the likelihood of cell ischemia and hypoxia. The anticancer mechanisms of saponins mostly involve interference with the proliferation of cancer cells by blocking specific test points of the cell cycle. Legume saponins block the cell cycle at the spindle test point during mitosis, directly interfering with the signalling pathway of cancer cell proliferation [35,36].

Soyasaponins I and III extracts can induce apoptosis of Hep-G2 cells (SKOV-3 and Saos-2 by activating caspases. In addition, soyasaponin II can induce HeLa cell apoptosis by increasing intracellular calcium, damaging mitochondrial functionality, and promoting the release of cytochrome C in the cytoplasm [37]. Extensive research has suggested that calcium dynamic equilibrium can be disrupted by saponins. The anticancer properties of saponins are based on inducing cell arrest during cell division cycle, which leads to cell apoptosis due to the imbalance of dynamic balance [29]. Saponins exert their anti-cancer abilities through anti-tumour activity, including caspases, protein kinase C (PKC), survivin and extracellular signal-regulated kinase (ERK), which inhibit a broader range of human cancers, including glioma, cervical cancer and colon cancer [29]. The cell cycle is divided into interphase and mitotic phase (M phase), and the interdivision phase is divided into three stages: DNA pre-synthetic phase (G1 phase), stage of DNA synthesis (S phase) and DNA post-synthetic phase (G2 phase). Two phases of the cell cycle are the most important: the G1 to S phase and the G2 to M phase, which is called cell cycle progression. These two stages are precisely the periods of complex and active molecular changes that are easily influenced by environmental conditions. The artificial regulation of

cell cycle progression is fundamental to the research on the development of organisms and controlling tumour growth. Cyclin and cyclin-dependent kinase (CDK) co-regulate cell cycle progression [38,39]. Different cyclins and their corresponding CDK complexes are responsible for diverse cell cycle transitions [28]. CDK binds to cyclins and CDK inhibitors (CDKI) and interacts to enhance the catalytic activity of CDK [29]. CDKI includes the INK (p15, p16, p18 and p19) and CIP/KIP families (p21, p27, p53 and p57). Overexpression of p21 in the CIP/KIP family results in cell cycle stagnation in the G1 phase. In addition, p53, an upstream regulator of p21, also damages DNA in the G1 phase and disrupts gene integrity [40]. However, the cell cycle progression hindrance of p21 and p53 is independent. Protein p21 obstructs cell cycle progression during the G2/M transition, while p53 obstructs cell proliferation during the G1 to S transition [41]. In addition, during the G2 to M transition phase, the cyclin B/cdc2 complex phosphorylates the nuclear medium and condenses chromatin thereby blocking mitosis [42].

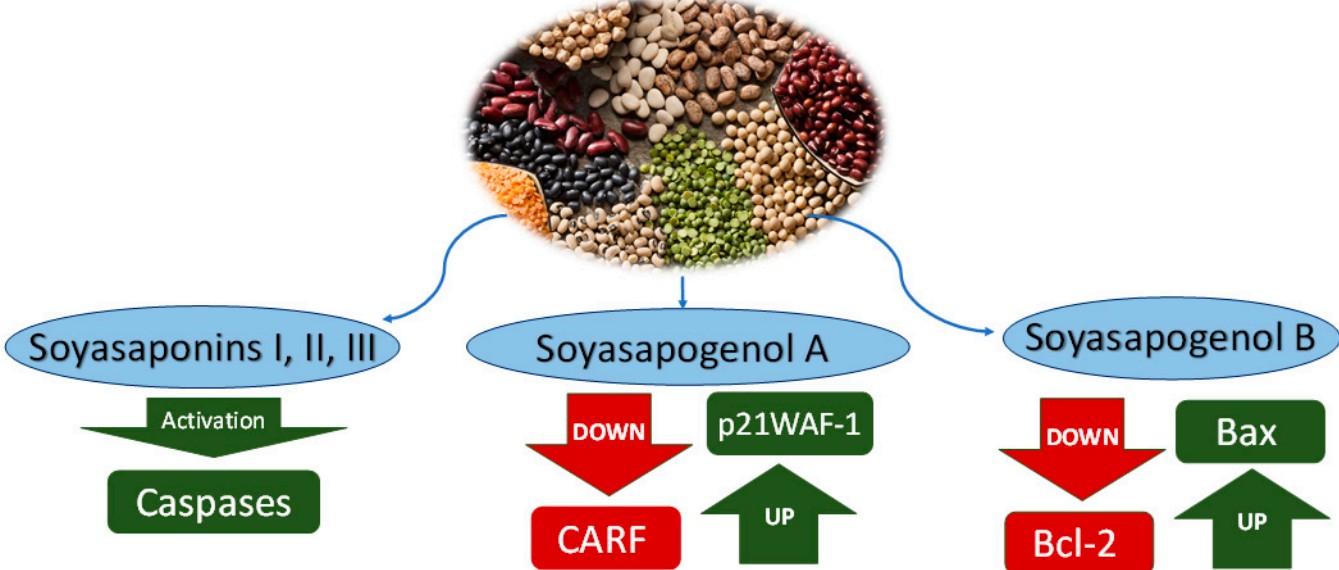

**Figure 2. Anticancer mechanisms of saponins from water extract of Leguminosae.** Soyasaponins I, II and III activate caspase enzymes. Soyasapogenol A downregulates CARF and upregulates p21WAF-1. Soyasapogenol B downregulates Bcl-2 and upregulates Bax.

It was found that soy saponin decreased the number of viable WiDr human colon cancer cells in a dose-dependent manner and suppressed PMA-induced PKC activity. Cells treated with saponins developed cytoplasmic vesicles with the cell membrane being rougher and more irregular in a dose-dependent manner and eventually disassembled. At 600 and 1200 ppm, the activity of AP was increased [43].

### 2.2.2. Soyasapogenol A

Soyasapogenol A and B extracted from leguminous such as faba, pea, chickpea, and soy are cytotoxic to liver cancer cells (Hep-G2), targeting the collaborator of ARF (CARF) protein in order to induce apoptosis or growth arrest [44] (Table 2). Data from several studies suggested that overexpression of CARF may lead to stagnation of cancer cell growth, as well as malignant transformation. CARF was found as an inherent potential therapeutic target for malignant tumours [45]. CARF is a vital protein for cell viability [46]. CARF is the significant basis of cell proliferation management through p53-HDM2 to p21 WAF-1 and DNA damage signalling 436]. It can play an anticancer mechanism through direct interaction with proteins such as p53, transcriptional inhibition of HMD2 and p21WAF-1, and epithelial–mesenchymal transition (EMT) promoting cancer cell invasion and malignant metastasis [35,37].

In addition, Snol-A is highly cytotoxic (50–70%) to other cancer cells: U2OS, wild-type p53, HT1080 fibrosarcoma (mutant p53), and breast cancer (McF-7; wild-type but inactive p53. In normal humans, fibrocytes appear milder (20–30% cytotoxic). The effect of Snol-A on osteosarcoma (U2OS; wild-type p53 and SAOS-2; Null p53), ovarian adenocarcinoma (SKOV3; Null p53) and breast adenocarcinoma (MDA-MB-231; p53 gene mutation) is repression of dose-dependent cell multiplication in all types of cancer cells. After the 48 h treatment of 2–10 μM Snol-A, U2OS and SKOV-3 cells presented stress phenotype as irregular and obese cell shapes, with genetic restriction compared to the normal group. The Snol-A-treated SKOV-3 and SAOS-2 cells lacking wild-type p53 function displayed considerable dose-dependent cytotoxicity as well [44–46].

Acting as a transcriptional inhibitor of p21WAF-1 in p53 deficient cells, CARF protein levels were obviously reduced in cells after the treatment of 10 μM Snol-A [34]. Snol-a-induced CARF inhibition was found to be like that expressed by p21WAF-1. Downregulation of CARF and upregulation of p21WAF-1 in SKOV-3 cells induced by Snol-A showed stronger effects at higher doses (over 6 μM) and over a longer treatment period (2 days) [47]. A similar result was achieved in p53 deficient Saos-2 cells and p53 mutant cells, which require higher doses of moderately toxic doses (IC50) compared with SKOV3 [44].

The suppression of the pATM-Chk1 signalling pathway can be caused by the Snol-A-mediated inhibition of CARF, which can promote cell apoptosis at higher concentrations. Overexpression of CARF caused activation of DNA damage response (DDR), which promoted growth arrest and ageing by activating the ATR-Chk1 pathway [34]. In Snol-A-treated cells, a decrease in dose-dependent CARF levels was accompanied by a reduction in ATR, pATR and Chk-1 expression. Furthermore, Snol-A-treated cells also showed a significant decrease in PARP1/2, while the corresponding lysis of PARP1/2 was elevated [44]. Consistent with these altered expression levels and acquired apoptosis phenotype, procaspase-9 and procaspase-3 decrease significantly in higher Snol-A concentrations treated cells, while the cleaved-caspase-3 increase in Snol-A-treated cells. Under the action of high doses of Snol-A, an increase in the expression of cleaved PARP1/2 and its nuclear staining confirmed the apoptosis of these cells.

Expression analysis of key proteins associated with cell migration and invasion signals showed decreased levels of CARF, β-catenin, vimentin, Smad 2/3, heterogeneous nuclear ribonucleic protein K (HNRNP-K) and matrix metalloproteinase-9 (MMP-9) proteins. Previous studies have shown that upregulation of CARF can promote nuclear enrichment of β-catenin. Immunostaining showed that β-catenin was significantly reduced in the nuclei of SKOV-3 treated with 2 and 4 μM Snol-A compared with the untreated control, suggesting that Snol-A-targeted CARF inhibition eliminated β-catenin nuclear function, leading to reduced cell migration and invasion. Immunostaining showed that the levels of vimentin and fibronectin, two key mesenchymal markers, decreased after 2 and 4 μM Snol-A treatment. The results showed that the levels of SMad2/3, HNRNP-K (A key effector protein involved in cell migration) and MMP- were significantly reduced after treatment with 2 and 4 μM Snol-A [44].

### 2.2.3. Soyasapogenol B

Soyasapogenol B (Snol-B) is an oleane-type pentacyclic triterpenoid and has wide applicability in food and health care [48]. Its biological activity is higher than other legume saponins, such as neuroprotective, anti-inflammatory and antiviral effects [48]. Moreover, research has demonstrated the anti-tumour effects of Snol-B in cancers such as breast cancer [49] and hepatocellular carcinoma [50]. Snol-B was found to induce anti-tumour effects and suppress the growth of laryngeal cancer cells HeP-2 and TU212 between the G0 and G1 cell cycle.

Snol-B has been shown to decrease the survival rate of Caco-2 cells. Results were observed upon administration of physiological doses of 0.15 mg/mL over incubation of 48–72 h [51]. The mechanism proposed was a reduced activity of protein kinase C (PKC)

### 2.3. Oligosaccharides from Leguminous and Anticancer Mechanisms

2.3.1. α Galactooligosaccharides (α-GOS)

Chickpea seeds (*Cicer arietinum*) are a significant source of α-linked galactooligosaccharide (α-GOS). Based on the dry mass, 10–17% of chickpeas are α-GOS. The most abundant α-GOS are ciceritol (36–43%) and stachyose (25%). It has been documented that α-GOS from soybeans are cytotoxic to breast cancer [52]. A recent in vivo study on mice has shown that oral intake of α-GOS at dietary dose of 2.2 g/kg body weight/day over 1 week reduced negative biomarkers associated to liver steatosis fasting glycemia, a disease that can lead to hepatocellular carcinoma (liver cancer) [52]. Specifically, blood levels of fasting glycemia, free fatty acids, low-density lipoprotein (LDL) and total cholesterol were reduced, well as the amount of food eaten. Meanwhile, no changes occurred to body weight, insulin sensitivity, high-density lipoprotein (HDL) and total triglycerides [52]. The α-GOS are considered prebiotic, which can promote the growth of probiotic microorganisms such as *Bifidobacterium* and the balance of the intestinal flora can reach the colon and serve as a substrate for the fermentation of the microbial flora [53]. Specifically, α-GOS are also metabolised by *Lactobacillus, Bifidobacterium* and *Lactobacillus*, stimulating the secretion of short-chain fatty acids, which in turn promote metabolism and relieve inflammation [53].

2.3.2. Raffinose and Stachyose

The raffinose family oligosaccharides (RFOs) are an important class of water-soluble carbohydrate, which have been demonstrated to help prevent diabetes and cardiovascular disease [54]. Oligosaccharides in the raffinose family (RFOs) are generally fermented by microorganisms in large plants to produce low concentrations of gases such as carbon dioxide, hydrogen and methane [55]. The composition of the intestinal microbiota has profound implications for nutrition, immunity, disease, and animal experiments showed that raffinose induced a decrease in the population of capsulated larvae in casein-fed rats and that the combination of soy protein and RFO produced cecal IgA concentrations much higher than those observed in other diets [56].

Raffinose is one of the most well-known trisaccharides in nature, consisting of galactose, fructose and glucose [57]. It is also a functional oligosaccharide with a strong *Bifidobacterium* proliferation effect. As an advanced prebiotic, RFO can protect the intestinal tract and reduce the risk of colon cancer [58].

Stachyose, a classical prebiotic in RFOs, is a functional oligosaccharide. It not only inhibits some intestinal diseases, such as ulcerative colitis but also selectively promotes the growth of intestinal probiotics [51,59]. The experiment showed that long-term high-fat diet (HFD)-induced inflammation in the colon and liver and imbalance of intestinal flora ecological balance could be improved by long-term intake of stachyose. The oligosaccharides raffinose and stachyose found in soybeans support *Bifidobacterium* proliferation [60], thereby reducing harmful bacteria and enzymes flora (such as azo reductase produced by carcinogens), maintaining intestinal health [61]. A large number of proliferating *Bifidobacterium* produce acetic acid, lactic acid and other organic acids during glycolysis of oligosaccharide, which can reduce the pH value of the intestinal cavity [62], thus accelerating intestinal propulsive movement, and forming defecation reaction to promote toxin discharge in the body [63] (Table 2; Figure 3).

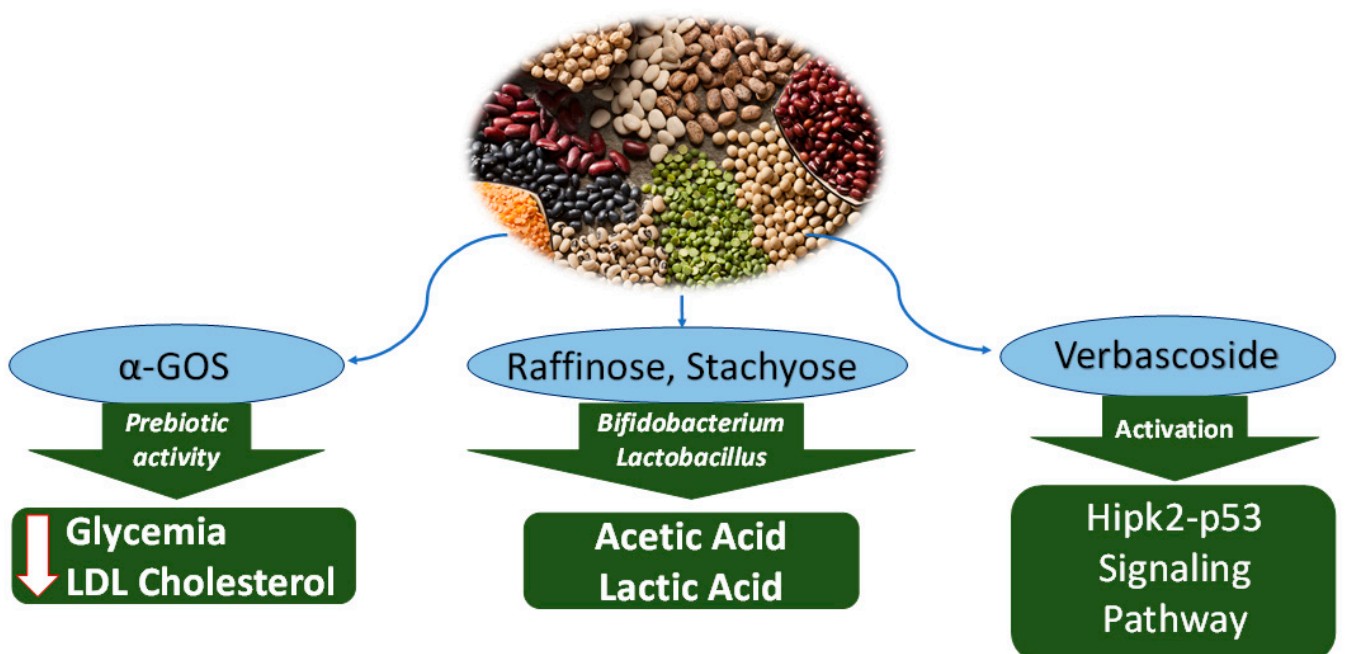

**Figure 3. Anticancer mechanisms of oligosaccharides from water extract of Leguminosae**. **A-GOS reduce glycemia and LDL-cholesterol.** Raffinose and stachyose are metabolised by probiotic bacteria, with synthesis of acetic acid and lactic acid, thus acidifying the intestinal pH. Verbascoside activates the signalling pathway of HIPk2-p53m causing apoptosis.

### 2.3.3. Verbascoside

Verbascoside is a pentose oligosaccharide. It has the function of proliferating beneficial bacteria, enhancing intestinal colonization resistance, inhibiting harmful bacteria, regulating intestinal microecology balance, and improving the intestinal tract [63,64]. The results showed that verbascoside could improve the activity of serum lysozyme and had strong immunostimulatory activity [65].

The main oligosaccharide in raw broad bean (*Vicia faba*) is verbascoside. The experiment showed that verbascoside (VB) had an inhibitory effect on colorectal cancer (CRC). HIPK2 samples and normal intestinal tissues from patients with primary colorectal cancer were analysed by HIPK2 analysis in vitro and in vivo. VB treatment significantly increased the expression of pro-apoptotic proteins and decreased the expression of anti-apoptotic factors in CRC cells. The results showed that VB effectively activated the Hipk2–p53 signalling pathway, leading to increased apoptosis in CRC cells [66] (Table 2). Aiming at the research progress of breast cancer treatment, the anticancer property of VB in T1 mouse breast tumour cells was studied by experiments. The results showed that verbascoside inhibited the proliferation of four kinds of T1 cancer cells [67]. Prostate cancer is a dangerous disease that leads to an increased mortality rate in men. It has been found through experiments that verbascoside can reduce the proliferation and invasion of prostate cancer cells. Verbascoside can become an effective nutritional supplement for patients with prostate cancer [68]. In addition, a study on breast cancer revealed the positive contribution of this oligosaccharide. Treatment of 4T1 cells revealed apoptosis. The mechanism observed was the activation of the Hipk2-p53 signalling pathway [61].

**Table 2.** Key anticancer mechanisms of soluble nutrients from Leguminosae.

| Nutrient | Source | Anticancer Effect | Anticancer Mechanism | Cell Type | Key References |
|---|---|---|---|---|---|
| **Bioactive Peptides and Protein** | | | | | |
| Albumin | *Cicer arietinum* (Chickpeas) | Anti-inflammatory | Reduction of nitric oxide levels Inhibition of proinflammatory cytokines (interferon IFN-γ, interleukins IL-6, IL-12) | RAW 264.7 | [21,22] |
| Bowman-Birk Inhibitor | *Glycine max* (Soybeans) | Antiproliferative of colon cancer | Inhibition of serine proteases | HT29 | [22] |
| Vicilin (Globulin polypeptide) | *Pisum sativum* (Peas) | Protection against obesity-associated metabolic disorders | Activation of peroxisome proliferator-activated receptor (PPAR-γ) | 3T3-L1 | [26] |
| **Saponins** | | | | | |
| Soyasaponins I and III | *Glycine max* (Soybeans) | Cytotoxicity to cancer cells | Activation of caspases leading to apoptosis | p53 cancer cell SKOV-3 and Saos-2 | [34] |
| Soyasaponin II | *Glycine max* (Soybeans) | Cytotoxicity to cancer cells | Increased intracellular calcium, damaged mitochondrial functionality and cytochrome C release, leading to apoptosis | HeLa | [27] |
| Soyasapogenol A | *Glycine max* (Soybeans) | Cytotoxicity to cancer cells | Downregulation of CARF and upregulation of p21WAF-1 inducing repression of cell multiplication (fibrosarcoma, osteosarcoma, ovarian adenocarcinoma, breast adenocarcinoma) | p53 cancer cell SKOV-3 and Saos-2 | [34] |
| Soyasapogenol B | *Glycine max* (Soybeans) | Cytotoxicity to human colon cancer cells | Inhibition of protein kinase C (PKC) | Caco-2 cells | [49] |
| **Oligosaccharides** | | | | | |
| α-linked galac-tooligosaccharide (α-GOS). | *Glycine max* (Soybeans) | Protection against obesity-associated metabolic disorders | Lower fasting glycemia, free fatty acids, low-density lipoprotein (LDL), total cholesterol | In vivo (Mice) | [52] |
| Raffinose, Stachyose | *Glycine max* (Soybeans) | Prebiotic | Proliferation of *Bifidobacterium* and *Lactobacillus*, leading to production of acetate, lactate and other organic acids, thus lowering intestinal pH | In vivo (Mice) | [59] |
| Verbascoside | | Cytotoxic to breast cancer | Activation of Hipk2-p53 signaling pathway, leading to apoptosis | *4T1* | [67] |

## 3. Conclusions

Aqueous extracts of Leguminosae are a complex mixture of soluble nutrients: albumin, globulin, oligosaccharides, phenolics and saponins. These organic compounds exert numerous bioactivities: anti-inflammatory, cytotoxic to cancer cells, antioxidants and antimicrobial. Relatively to cancer, aqueous extracts of common beans, chickpeas, lentils, mung beans, peas and soybeans can express preventive effects via a plethora of mechanisms. Albumin from chickpeas can inhibit proinflammatory cytokines, whereas the BBI from soybeans inhibited serine proteases and vicilin from peas activated PPAR-γ. Soyasaponins are cytotoxic to numerous cancer cells via the activation of caspases and downregulation of Bcl-2. Oligosaccharides act as prebiotics and are cytotoxic to breast cancer cells. These findings highlight the potential application of water solutions from legumes. For example, upcycled water from legume processing (soaking, blanching, boil-

ing) is an aqueous extract of Leguminosae that should be investigated as a nutraceutical in the prevention and treatment of cancers.

**Author Contributions:** Conceptualization, L.S.; methodology, L.S.; software, L.S.; validation, L.S.; formal analysis, L.S.; investigation, L.S.; resources, L.S.; data curation, L.S.; writing—original draft preparation, L.S., X.C., R.C., N.D., J.S., R.P.N.T., E.E.H.; writing—review and editing, L.S.; visualization, L.S.; supervision, L.S.; project administration, L.S.; funding acquisition, L.S. All authors have read and agreed to the published version of the manuscript.

**Funding:** The APC was funded by the Lincoln University Open Access Fund.

**Institutional Review Board Statement:** Not applicable.

**Informed Consent Statement:** Not applicable.

**Data Availability Statement:** Not applicable.

**Acknowledgments:** Authors thank Lincoln University for the resources available to the Open Access Fund.

**Conflicts of Interest:** The authors declare no conflict of interest.

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
