# Peer review of "Anticancer Properties of Aqueous Extracts from Leguminosae"

_nutraceuticals, doi:10.3390/nutraceuticals2040025_

Round 1

Reviewer 1 Report

Comments and Suggestions for Authors

1.    Reduce title to: “Anticancer properties of aqueous extracts from Leguminosae

2.    Include scientific names in the abstract and throughout the review when appropriate.

3.    Use the term “leguminous” instead of “legume” or “legumes” to avoid confusion with other vegetables.

4.    Several biochemical and molecular nomenclature issues need to be reviewed and corrected. For instance, but not only: albumin is not just “a peptide” (as indicated in table 1), but a polypeptide, or more properly a protein. In addition, is it “vicilin” or “vicillin”? (with one or two "l "s?)

5.    A PubMed search on the terms: “anticancer leguminosae aqueous extract” yields 48 articles. It would be necessary to check to see if all relevant information is included here.

6.    The names of the molecules included in the text, table and figures must coincide. There are some differences, for instance: Globulin is not included in the table or in the figure, and vicilin does not have a specific section in the text, but is included in the table and figures. Soyasaponins are mentioned as I, II and III in the text and table, but not in the figure (and the order of mention is changing). Alpha-GOS is mentioned in the text but is not included in the table or figure.

7.    It should be useful to make a Table indicating the main effects or activities of the different molecules obtained from Leguminosae and present in the aqueous extract, for instance: anti-inflammatory, antioxidant, antiproliferative, cytotoxic, and others. Indicating in which cell type the effects where observed.

8.    The text includes some relevant experimental information; however, the redaction is obscure and confusing. For instance, but not solely: Line 42; “Consequently, studies about the tumor mechanism can be approached in two ways.”  which two ways? Line 86, it is necessary to explain: “nitric oxide was compared…” Where? Why? How? Line 91, explain: “investigated albumin extract from pea”… the effect of? And so on…

9.    Table 1. it should be included the cell type in which the action or effect was observed and whether it was in vitro or in vivo. In addition, references may be indicated by number, instead of last name and year, to reduce the width in the last column.

10. Figures 1 to 3 Should increase their quality and complexity. Delete the background color. Suppress intense and diverse colors.

11. Line 99 the first time BBI is mentioned should be explained.

12. Line 105 What “all aforementioned cytokines”?

Author Response

  1. Reduce title to: “Anticancer properties of aqueous extracts from Leguminosae”

The title was edited accordingly

  1. Include scientific names in the abstract and throughout the review when appropriate.

Scientific names were included as suggested

  1. Use the term “leguminous” instead of “legume” or “legumes” to avoid confusion with other vegetables.

The change has been made

  1. Several biochemical and molecular nomenclature issues need to be reviewed and corrected. For instance, but not only: albumin is not just “a peptide” (as indicated in table 1), but a polypeptide, or more properly a protein. In addition, is it “vicilin” or “vicillin”? (with one or two "l "s?)

Nomenclature issues have been corrected. Albumin is a polypeptide. Vicilin is the correct term.

  1. A PubMed search on the terms: “anticancer leguminosae aqueous extract” yields 48 articles. It would be necessary to check to see if all relevant information is included here.

Thanks for the suggestion. Our research focused on the past 5 years (recent findings). This search on PubMed resulted in 27 manuscripts. Some of these papers are not on leguminosae, but other seeds such as fenugreek.

  1. The names of the molecules included in the text, table and figures must coincide. There are some differences, for instance: Globulin is not included in the table or in the figure, and vicilin does not have a specific section in the text, but is included in the table and figures. Soyasaponins are mentioned as I, II and III in the text and table, but not in the figure (and the order of mention is changing). Alpha-GOS is mentioned in the text but is not included in the table or figure.

Table and text were edited to express representative lists of all bioactives discussed. Globulin weas included in the table as source of Vicilin. Vicilin was given a specific section (title change) in the text. Soyasaponins I, II and II were mentioned to Figure 2, keeping the same order of mention of the text. Alpha-GOS is included in the table and in Figure 3.

  1. It should be useful to make a Table indicating the main effects or activities of the different molecules obtained from Leguminosae and present in the aqueous extract, for instance: anti-inflammatory, antioxidant, antiproliferative, cytotoxic, and others. Indicating in which cell type the effects where observed.

A new table was added (named Table 1) listing the key nutrients of water extracts from Leguminosae, with their source, activity, cell type and references.

  1. The text includes some relevant experimental information; however, the redaction is obscure and confusing. For instance, but not solely: Line 42; “Consequently, studies about the tumor mechanism can be approached in two ways.” which two ways? Line 86, it is necessary to explain: “nitric oxide was compared…” Where? Why? How? Line 91, explain: “investigated albumin extract from pea”… the effect of? And so on…

Lines 45-62 have been deleted.

Line 86: details have been added, nitric oxide levels were compared between sprouted and non-sprouted chickpeas.

Line 91 has been implemented: “…investigated the effect of albumin…”

  1. Table 1. it should be included the cell type in which the action or effect was observed and whether it was in vitro or in vivo. In addition, references may be indicated by number, instead of last name and year, to reduce the width in the last column.

Table 1 was renamed Table 2. Source, effect and cell type were added. References were listed as numbers.

  1. Figures 1 to 3 Should increase their quality and complexity. Delete the background color. Suppress intense and diverse colors.

Background colour was deleted. Other colours were harmonized.

  1. Line 99 the first time BBI is mentioned should be explained.

BBI was spelled out to Bowman-Birk Inhibitor.

  1. Line 105 What “all aforementioned cytokines”?

The word “aforementioned” was removed due to changes in the paragraph structure, Details were added: cytokines mentioned are interferon IFN-É£, interleukins IL-6, IL-12.

Reviewer 2 Report

Comments and Suggestions for Authors

The authors highlighted the anticancer properties and the mechanisms of action of aqueous extracts from chickpeas, faba beans, peas and soybeans. In general, the review is well-written and well-organized. It can be accepted for publication upon following revision:

1. In the manuscript it was noted that a lot of reviews were referenced, it is
advised that the authors cite the original article(s) of the discovery.

2. The figures should be re-designed with the help of some tools to be more representable.

3. In Table 1, the authors should re-organize and separate mechanisms (e.g. downregulation of protein Bcl-2, upregulation of Bax) and effect (e.g. increased apoptosis) for better understanding.

4. The authors should aware of the use of proper latin alphabet e.g. PPARα instead of PPAR-alpha.

Author Response

Reviewer 2 

The authors highlighted the anticancer properties and the mechanisms of action of aqueous extracts from chickpeas, faba beans, peas and soybeans. In general, the review is well-written and well-organized. It can be accepted for publication upon following revision: 

On behalf of all co-authors, I thank the reviewer for the constructive feedback. 

  1. In the manuscript it was noted that a lot of reviews were referenced, it is

advised that the authors cite the original article(s) of the discovery. 

Specific research papers have been introduced. 

  1. The figures should be re-designed with the help of some tools to be more representable.

Figures have been re-designed to improve visibility. 

  1. In Table 1, the authors should re-organize and separate mechanisms (e.g. downregulation of protein Bcl-2, upregulation of Bax) and effect (e.g. increased apoptosis) for better understanding.

Table 1 was renamed Table 2 and edited accordingly, adding source and cell type, as well as separating effect and mechanisms. 

  1. The authors should aware of the use of proper latin alphabet e.g. PPARα instead of PPAR-alpha.

Proper Latin alphabet has been inserted. 

Reviewer 3 Report

Comments and Suggestions for Authors

Title: “Anticancer properties of aqueous extracts from chickpeas, faba beans, peas and soybeans

Authors: Luca Serventi, Xuanyi Cai, Ruitian Chen, Nadeesha Dilrukshi,
Jingyi Su, Refi Priskila Novaleta Tuange, And Elizabeth Eilidh Ham

Comments:

In this review paper, the authors would like to present the anti-cancer effects of plant water extracts. I find it positive that at the beginning of the paper the connection between environmental factors, inflammation and cancer is addressed. Within the paper, the focus on the main topic is lost. Overall, the review paper is not very informative, so both the content and style should be improved.

Examples:

1) Introduction: the first paragraph is good, it should be added if the statement in line 39/40 applies to all cancers and which two pathways are meant in line 43. Lines 44-66 do not fit the title topic, sentence transitions are missing here and the focus cancer is lost.

2) Also in the rest of the text it is difficult for the reader to identify the topic focus, as there are whole chapters where cancer is not mentioned, e.g. "albumin", "globulin", "α-GOS".

3) Table 1 should be extended with some columns (herbal sources containing the substances mentioned / are the references vitro or vivo experiments / which cancer type, which cells and which doses were studied) to have a useful information value.

4) In all chapters, more should be said about the study results on cancer and their significance. The conclusion should state how the results might be relevant to prevention. When cell lines are mentioned, it should be explained which cells are involved, e.g., line 24, 25.

5) Literature references should be revised. In some places source citations are not given or not linked correctly, e.g. line 99, 111, 121, 127. A plagiarism check should be done, e.g. line 70-79 is partially cited and line 80-110 is fully cited from ref 34.

6) Care should be taken to write out abbreviations when they are first used, for example line 52, 99, 109.

Author Response

Title: “Anticancer properties of aqueous extracts from chickpeas, faba beans, peas and soybeans”

Authors: Luca Serventi, Xuanyi Cai, Ruitian Chen, Nadeesha Dilrukshi,

Jingyi Su, Refi Priskila Novaleta Tuange, And Elizabeth Eilidh Ham

Comments:

In this review paper, the authors would like to present the anti-cancer effects of plant water extracts. I find it positive that at the beginning of the paper the connection between environmental factors, inflammation and cancer is addressed. Within the paper, the focus on the main topic is lost. Overall, the review paper is not very informative, so both the content and style should be improved.

The feedback is appreciated. Apologies for the lost of main topic over the main text. Therefore, the manuscript has been edited, as well as tables and figures, to provide a logic flow of information, addressing the research question.

Examples:

1) Introduction: the first paragraph is good, it should be added if the statement in line 39/40 applies to all cancers and which two pathways are meant in line 43. Lines 44-66 do not fit the title topic, sentence transitions are missing here and the focus cancer is lost.

The paragraph in lines 44-62 is not directly focusing on cancer, but rather on prebiotics. Therefore, it’s agreed to remove it.

2) Also in the rest of the text it is difficult for the reader to identify the topic focus, as there are whole chapters where cancer is not mentioned, e.g. "albumin", "globulin", "α-GOS".

A focus on cancer has been added to the chapters “Albumin”, “Vicilin” and "α-GOS".

3) Table 1 should be extended with some columns (herbal sources containing the substances mentioned / are the references vitro or vivo experiments / which cancer type, which cells and which doses were studied) to have a useful information value.

Source, effect and cell type were added.

4) In all chapters, more should be said about the study results on cancer and their significance. The conclusion should state how the results might be relevant to prevention. When cell lines are mentioned, it should be explained which cells are involved, e.g., line 24, 25.

Specific focus on cancer has been added to all chapters.

Conclusion was re-written

Cell lines are detailed: Lines 93, 114, 130, 170-171, 267-274

5) Literature references should be revised. In some places source citations are not given or not linked correctly, e.g. line 99, 111, 121, 127. A plagiarism check should be done, e.g. line 70-79 is partially cited and line 80-110 is fully cited from ref 34.

References have been revised, specifically in lines 111,121,127.

Lines 70-77 cite 3 manuscripts.

Lines 80-100 have been re-written. “Another study investigated the effect of albumin extract from pea against inflammation of mouse colitis [12]. Colitis was induced by adding Dextrane Sulfate Sodium in drinking water over the course of four days. The colitic mice treated with the aqueous pea extract showed a reduced mRNA expression of cytokines TNF-α, IFN-γ, IL-6 and IL-12.Despite studies conducted have investigated the potential anti-inflammatory property of albumin protein, investigation of other bioactive components such as the Bowman-Birk Inhibitor (BBI), antioxidants flavones cannot be excluded.”

6) Care should be taken to write out abbreviations when they are first used, for example line 52, 99, 109.

Line 52 has been deleted. Lines 99 and 109 have been implemented with the explanation for BBI (Bowman-Birk Inhibitor).

Round 2

Reviewer 1 Report

Comments and Suggestions for Authors

The authors have addressed most of the questions and comments, consequently the manuscript has been improved in precision.

Reviewer 3 Report

Comments and Suggestions for Authors

The authors have satisfactorily addressed the concerns raised in the original version. The revised version is significantly improved. No further concerns.